

# Metal tolerance capacity and antioxidant responses of new *Salix spp.* clones in a combined Cd-Pb polluted system

Chuanfeng Zhang[1,*], Baoshan Yang[1,*], Hui Wang[1,2,3], Xiaohan Xu[4], Jiaxing Shi[1] and Guanghua Qin[5]

[1] School of Water Conservancy and Environment, University of Jinan, Jinan, Shandong province, China
[2] Shaanxi Key Laboratory of Land Consolidation, Xi'an, Chian
[3] Chang'an University, Xi'an, China
[4] College of Horticulture, Nanjing Agricultural University, Nanjing, Jiangsu province, China
[5] Shandong Academy of Forestry, Jinan, Shandong province, China
[*] These authors contributed equally to this work.

Corresponding author
Hui Wang, ecogroup@126.com

## ABSTRACT

To investigate the physiochemical characteristics of two new clones, *Salix matsudana* 'J172' (A7) and *Salix matsudana* 'Yankang1' (A64) in combined Cd-Pb contaminated systems, a hydroponic experiment was designed. The plant biomass, photosynthesis, antioxidant responses and the accumulation of metals in different plant parts (leaf, stem, and root) were measured after 35-day treatments with Cd (15, 30 $\mu$M) and Pb (250, 500 $\mu$M). The results showed that exposure to Cd-Pb decreased the biomass but increased the net photosynthetic rate for both A7 and A64, demonstrating that photosynthesis may be one of the metabolic processes used to resist Cd-Pb stress. Compared with control, roots exposed to Cd-Pb had higher activity of superoxide dismutase and more malondialdehyde concentrations, which indicated the roots of both clones were apt to be damaged. The concentrations of soluble protein were obviously higher in the roots of A64 than A7, indicating the roles of the antioxidative substance were different between two willow clones. Soluble protein also had significant relationship with translocation factors from accumulation in roots of A64, which illustrated it played important roles in the tolerance of A64 roots to heavy metals. The roots could accumulate more Pb rather than transport to the shoots compared with Cd. The tolerance index was more than 85% on average for both clones under all the treatments, indicating their tolerance capacities to the combined stress of Cd and Pb are strong under the tested metal levels. Both clones are the good candidates for phytoremediation of Cd and Pb by the root filtration in the combined contamination environment.

## INTRODUCTION

A large quantity of heavy metals (HMs) as the potentially toxic elements was released into environment with the rapid development of industry and agriculture throughout the world, which has posed a serious threat to the human being and biota (*Liao et al., 2015*; *Zhou et al., 2019*; *Zuzolo et al., 2022*). Soil and aquatic environment have been contaminated by

multiple HMs due to mining, smelting, dumpling of solid wastes and so on (*Wang et al., 2016*). Cadmium (Cd) and Lead (Pb) are considered as hazardous contaminants because of their high toxicity, mobility, and non-biodegradable (*Chen et al., 2013*; *Han et al., 2021*). Co-occurrence of Cd and Pb is pervasive in the environment, particularly in waters and soil. A number of adverse impacts of Cd and Pb contamination on plants, animals and microorganisms have been reported (*Xu et al., 2019*; *Zhou et al., 2019*), for example, the noxious effects of Cd and Pb on growth, nutrient uptake and oxidative damages in plants have been confirmed (*Xin et al., 2010*; *Zulfiqar et al., 2019*). The cost-effective remediation technology was widely explored in the environment contaminated by Cd and Pb.

Various physical, chemical and biological techniques have been exploited to remove HMs from environment (*Sarwar et al., 2017*). Phytoremediation is an eco-friendly and effective method, which can remediate the contaminated environment by phytoextraction and phytostabilization with little disturbance to the environment (*Ali, Khan & Sajad, 2013*; *Lee et al., 2019*). For example, *Bai et al. (2022)* found elm/poplar decreased the contents of HMs in the remediation areas after ten years of phytoremediation. In addition, *Festuca*, as one plant with overexpression of metal transporters, could recovery the environment contaminated by potentially toxic elements (*Zuzolo et al., 2022*). HMs can disturb chlorophyll biosynthesis and lower mineral uptake, leading to plant growth inhibition even necrosis (Haider et al., 2021). Therefore, plant accumulation and tolerance capacity to HMs has been regarded as the most common screening emphasis in the phytoremediation of HMs-contaminated sites.

Plant accumulation and tolerance capacities are regulated by various adaptive mechanisms, including the role of transporters in HMs accumulation, antioxidant defenses to maintain cellular redox homeostasis, detoxification mediated by phytochelatins and metallothioneins (*Raza et al., 2020*). In response to Cd/Pb oxidative stress, plants could produce various types of enzymatic and non-enzymatic antioxidants such as superoxide dismutase (SOD), peroxidase (POD), catalase (CAT) and free prolines to lessen the noxious effects of reactive oxygen species (ROS) (*Zulfiqar et al., 2019*). The previous studies found that POD, SOD, and CAT activities in *Phaseolus vulgaris* were increased with the increasing Cd and Pb concentrations (*Hammami et al., 2022*).

Willows have been paid widespread attention in the remediation of contaminated environment with HMs owing to their strong HMs accumulation, high biomass and deep root system (*Fortin Faubert et al., 2021*; *Gervais-Bergeron, Chagnon & Labrecque, 2021*; *Meers et al., 2007*). Willows could grow in contaminated environments due to the sufficient tolerance, and the obtained biomass might subsequently be used for bio-fuel production (*Meers et al., 2007*). Previous studies have found either Cd or Pb caused significantly negative effects on the biomass of leaves and roots for willows (*Xu et al., 2019*). The stress of Cd or Pb could significantly increase the activities of SOD and malondialdehyde (MDA) contents, while suppressed the activities of CAT and POD in leaves of *Salix matsudana* Koidz. 'Shidi1' and *Salix psammophila* C. 'Huangpi1' (*Xu et al., 2019*). The study of *Wang et al. (2019)* indicated that the roots of *S. matsudana* enhanced the tolerance and extraction capacity in Cd and Cu treatments. When compared among 12 willow clones about the tolerance to Cu and Zn, *Yang et al. (2014)* found that the tolerance
and accumulation in willows varied with clones. Although the phytoremediation of Cd and Pb has been extensively investigated, there is little comprehensive information on the tolerance characteristics and antioxidant responses of the new breeding willow clones in the combined contamination systems with different Cd and Pb concentrations. It is necessary that the wide ranges of willow clones are explored for the implement of phytoremediation in the Cd and Pb contaminated environment.

We carried out a hydroponic experiment in the system of co-occurrence of Cd and Pb in this study, which are easier to control the concentration of Cd and Pb compared with the soil system. The objectives are to investigate: (1) phytoextraction capacities of Cd and Pb in the two new breeding willow clones of *Salix matsudana* 'J172' (A7) and *Salix matsudana* 'Yankang1' (A64); (2) the growth, photosynthesis and antioxidant mechanism of two willow clones under the joint pollution of Cd and Pb; (3) the correlation between HMs phytoextraction and tolerance and physiochemical responses of two willow clones. These results will provide a base for the screen of the functional plant in the phytoremediation of HMs contaminated environment.

## MATERIALS & METHODS

### Plant material and experimental design

The one-year-old branches of *Salix matsudana* 'J172' (A7) and *Salix matsudana* 'Yankang1' (A64) were collected from experimental sites of Shandong academy of forestry in Shandong Province, China. The cuttings with a length of 20 cm and a diameter of 1.5–2.5 cm were pretreated according to the disinfection methods of *Xu et al. (2019)*, then cultivated for three weeks in the water in glass jar for rooting. Then they were transferred to the culture tube containing 4 L of 100% Hoagland nutrient solution, which was replaced once a week. All the cuttings were cultivated in the greenhouse in University of Jinan (Jinan, China) under the natural light, relative humidity 75% $\pm$ 10%, and temperature 24−30 °C. Three weeks after the cultivation, the seedlings developed mature roots and new branch sprout (*Xu et al., 2019*), and the plants with consistent height and diameter were selected for the following HMs stress. Five treatments were applied: 0 (CK) and combined treatments Cd (CdCl$_2$ 5/2H$_2$O, 15 and 30 $\mu$M) and Pb (PbCl$_2$, 250 and 500 $\mu$M) with four replicates. The corresponding Cd and Pb treatment groups were marked as CK, LCdLPb, HCdLPb, LCdHPb, and HCdHPb (Table S1). We kept the water volume by the supplementation of distilled water. The HMs concentrations were maintained *via* replacing the nutrient solution every two weeks. The stress experiment lasted five weeks before the plants were harvested.

### Determination of photosynthetic parameters and oxidative indicators

Before one day of harvest, the leaves of three plants from each treatment were selected for the determination of photosynthetic parameters, including net photosynthetic rate (*Pn*), stomatal conductance (*Gs*), intercellular CO$_2$ concentration (*Ci*) and transpiration (*Tr*), using a full-automatic portable photosynthesis system (LC pro-SD; ADC, Hoddesdon, UK). The harvested cuttings were separated into leaves, stems (including barks and woody cuttings), and roots. Antioxidant enzymes, antioxidant substances and MDA were

determined for each portion of the fresh plant samples. The other residual tissues were dried at 85 °C for the determination of dry weight and HMs content.

To compare the resistance of new breeding willow clones to the combined stress of Cd and Pb, we calculated the dimensionless sensitivity index (SI) by Eq. (1) (*Xin et al., 2010*).

$$SI\ (\%) = (DW_{control} - DW_{HMs})/DW_{control} \times 100 \tag{1}$$

where $DW_{HMs}$ is the dry weight of the plant under the combined Cd and Pb treatments, and $DW_{control}$ is the dry weight of the plant under control.

The physiological parameters were measured after grinding the frozen fresh plant organs with liquid nitrogen. The contents of MDA, free proline and soluble protein were determined according to the methods of *Lei, Korpelainen & Li (2007)*, *Tamás et al. (2008)* and *Zheng, Fei & Huang (2009)*. The activities of POD, CAT and SOD were estimated using the methods described by *Xu et al. (2019)*.

## Cd and Pb analysis in plant

Plant samples (0.1 g) were oven-dried, pulverized by grinding with mortar manually. Mixed acid ($HNO_3$-$HClO_4$) was used to digest samples according to the methods of *Xu et al. (2019)*. The concentrations of Cd and Pb in the samples were determined by atomic absorption spectroscopy (SHIMADIV AA-7000, Japan).

Bioconcentration factor (BCF) and translocation factors from concentration (TF) and accumulation (TF') were used to quantify the efficiency of plant extraction of HMs. TF and TF' also indicate the ability of plants to translocate Cd and Pb from the roots to the shoots. The formula was presented as follows:

$$BCF = C_{plant}/C_{solution} \tag{2}$$

where $C_{plant}$ is the concentration of HMs in the harvested plant tissue and $C_{solution}$ is the concentration of the same metal in the solution.

$$TF = C_{shoot}/C_{root} \tag{3}$$

where $C_{shoot}$ is the HMs concentration in the plant shoot and $C_{root}$ is the HMs concentration in the plant root.

$$TF' = M_{shoot}/M_{root} \tag{4}$$

where $M_{shoot}$ and $M_{root}$ represent the HMs content in plant shoots and roots, which is calculated by shoot (root) biomass × metal concentration.

The tolerance index (Ti) was calculated to evaluate the ability of the two willow clones to grow under the metal treatments (*Wu et al., 2010*; *Xu et al., 2019*):

$$Ti = DW_{HMs}/DW_{control} \tag{5}$$

$DW_{HMs}$ and $DW_{control}$ are the dry weight of plant under HMs treatment and control, respectively.
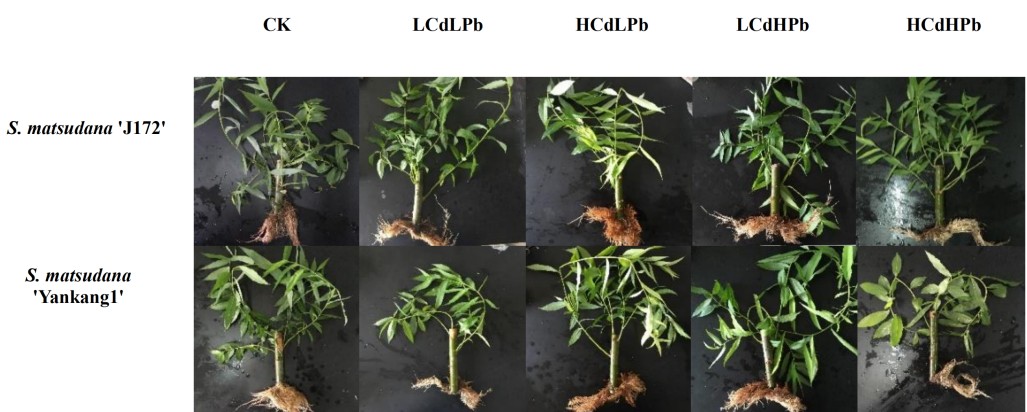

|  | CK | LCdLPb | HCdLPb | LCdHPb | HCdHPb |
| --- | --- | --- | --- | --- | --- |

*S. matsudana* 'J172'

*S. matsudana* 'Yankang1'

**Figure 1** **The phenotypic response of *S. matsudana* 'J172' and *S. matsudana* 'Yankang1' to control and the combined Cd and Pb stress.**

## Statistical analysis

All the figures were performed using Origin (version 9.8.0.200). The statistical analysis was performed using the SPSS software (version 19.0, SPSS Inc., Chicago, IL, USA)) according to LSD test. The significant difference among treatments was considered at $P < 0.05$. For the mantel analysis, metal tolerance capacities (Ti, BCF, TF and TF') were transformed into Bray distances, and other variables used Euclidean distances.

## RESULTS

### The effects of combined stress of Cd and Pb on plant biomass

After 35 days, the combined exposure of Cd and Pb significantly affected the biomass of two *S. matsudana* clones but did not reveal the signs of overt stress (Figs. 1–2). Compared with control, the decrease of the leaf and root biomass of A7 and A64 was observed under each HMs treatment, while the stem biomass was hardly changed (Fig. 2). The leaf biomass decreased by 10% in both A7 and A64, but root biomass decreased by 67% and 64%, respectively, indicating the root biomass was more susceptible to Cd and Pb stress (Fig. 2). Figure 3 also showed the largest inhibition appeared in roots with the range from 60% to 80% compared to the leaves and stems both in A7 and A64. The stems suffered the lowest inhibition under the combined stress of Cd and Pb (Fig. 3). Most of the leaf and root biomass of A64 was significantly higher than that of A7 except for the value under HCdHPb (Fig. 2).

Sensitivity index (SI) varied with the willow clones, organs and treatments (Fig. 3). Combined stress of Cd and Pb caused inconsistent SI on A7 and A64. The stronger negative impact of Cd and Pb on the leaves was found in A64 rather than A7 under the treatments of LCdLPb and HCdHPb, while HCdLPb and LCdHPb caused the strong inhibition on the leaves of A7 rather than A64. It is worth noting that HCdHPb did cause the strongest effects on the leaves and roots of two willow clones except for the suppression on the leaves of A7.

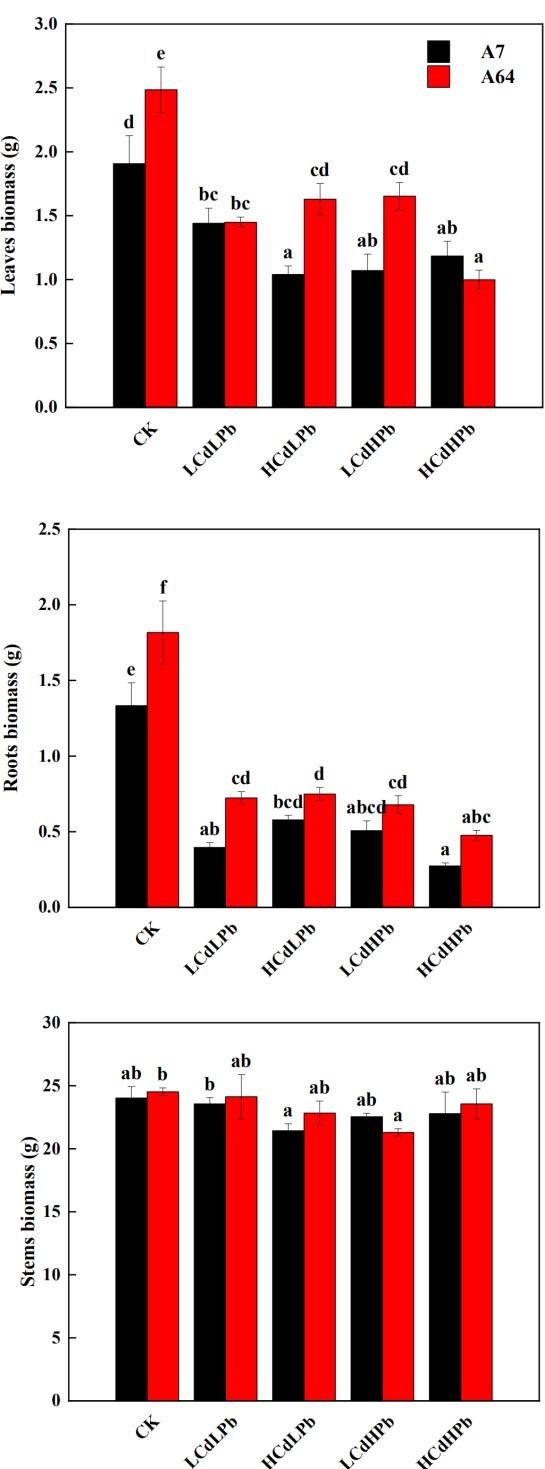

**Figure 2** **Dry biomass of leaves, roots and stems for *S. matsudana* 'J172' (A7) and *S. matsudana* 'Yankang1' (A64) under the combined Cd and Pb stress.** CK, LCdLPb, HCdLPb, LCdHPb and HCdHPb present no added metals, 15 μM CdCl$_2$ + 250 μM PbCl$_2$, 30 μM CdCl$_2$ + 250 μM PbCl$_2$, 15 μM CdCl$_2$ + 500 μM PbCl$_2$ and 30 μM CdCl$_2$ + 500 μM PbCl$_2$. Each value represents the mean of the four plants ± SD. Different lowercase letters indicate significant differences among different treatments at the $P < 0.05$ level, as determined by LSD test.

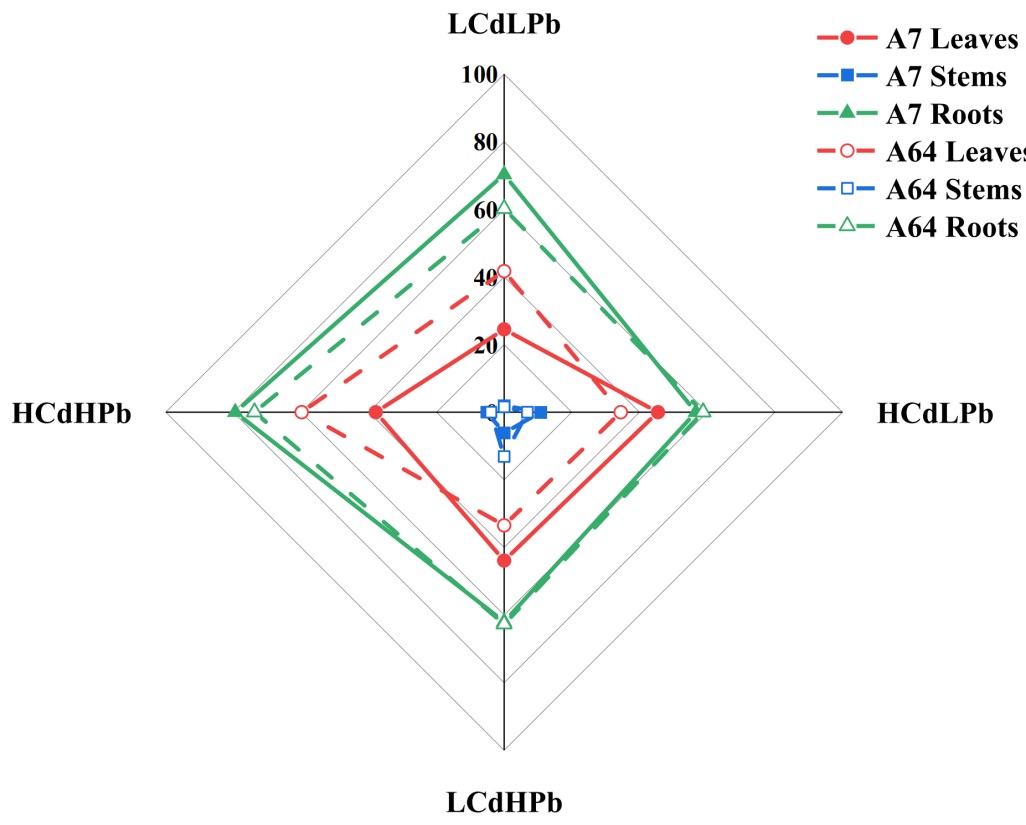

**Figure 3** **Sensitivity index (SI) for *S. matsudana* 'J172' (A7) and *S. matsudana* 'Yankang1' (A64) under the combined Cd and Pb stress.** CK, LCdLPb, HCdLPb, LCdHPb and HCdHPb present no added metals, 15 μM $CdCl_2$ + 250 μM $PbCl_2$, 30 μM $CdCl_2$ + 250 μM $PbCl_2$, 15 μM $CdCl_2$ + 500 μM $PbCl_2$ and 30 μM $CdCl_2$ + 500 μM $PbCl_2$.

## The effects of combined stress of Cd and Pb on plant photosynthesis

The photosynthetic responses of the two *S. matsudana* clones to different concentrations of Cd and Pb were remarkably different (Fig. 4). *Pn* of A7 was significantly higher than that of A64 under all treatments. Compared with the control, all the treatments increased *Pn* of A7, especially for HCdLPb. It's worth noting that *Pn* of A64 was increased by all treatments, although the values in HCdLPb were not significant. The *Gs* of A7 was significantly higher than that of A64 except for HCdHPb. Under the low Pb treatment (LCdLPb and HCdLPb), the *Gs* values of both A7 and A64 were higher than the control. However, the high Pb concentrations (LCdHPb and HCdHPb) declined the *Gs* values of two willow clones. In addition, *Ci* values of the two clones had a slight decrease and the differences were not significant except for HCdHPb. Interestingly, under the treatments of CK, HCdLPb and LCdHPb, the *Tr* values of A7 were significantly higher than those of A64, while the *Tr* of A64 was higher than A7 under the treatment of HCdHPb.

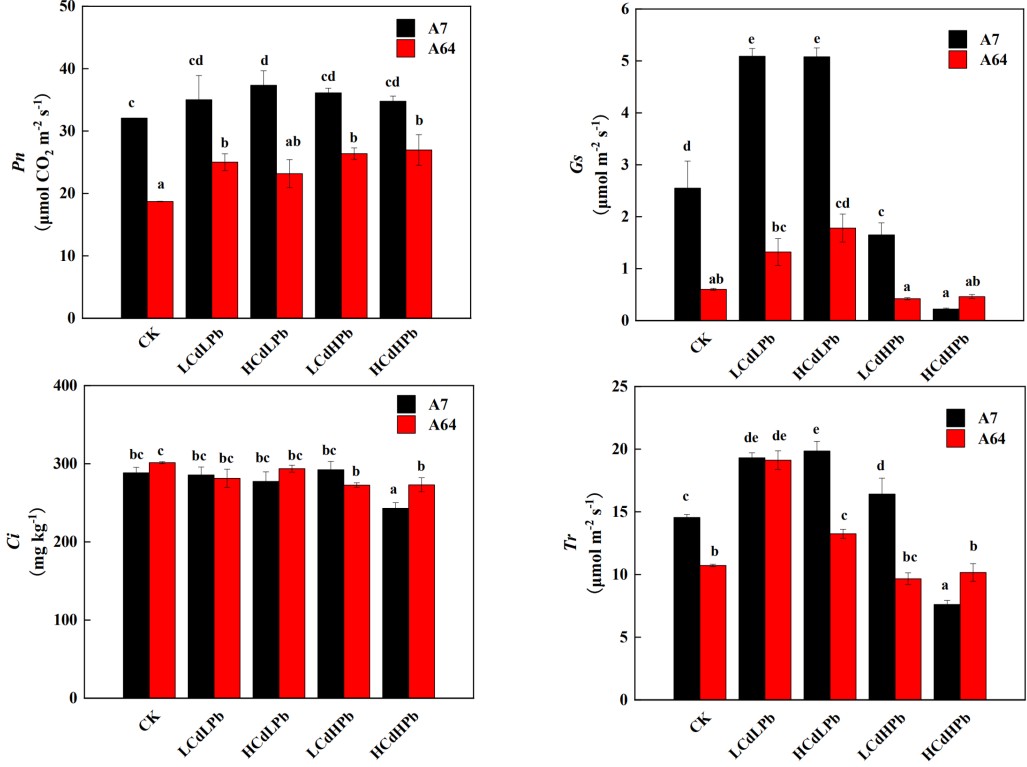

**Figure 4  Net photosynthetic rate (*Pn*), stomatal conductance (*Gs*), intercellular CO₂ mole fraction (*Ci*) and transpiration (*Tr*) in leaves of A7 and A64 treated with combined Cd and Pb.** A7 and A64 present *S. matsudana* 'J172' and *S. matsudana* 'Yankang1', respectively. CK, LCdLPb, HCdLPb, LCdHPb and HCdHPb present no added metals, 15 μM CdCl₂+ 250 μM PbCl₂, 30 μM CdCl₂+ 250 μM PbCl₂, 15 μM CdCl₂+ 500 μM PbCl₂ and 30 μM CdCl₂+ 500 μM PbCl₂. Each value represents the mean of the four plants ± SD. Different lowercase letters indicate significant differences among different treatments at the *P* < 0.05 level, as determined by LSD test.

## The effects of combined stress of Cd and Pb on plant antioxidant system

Figure 5 indicated that MDA contents significantly decreased in leaves of A7 and A64 under the combined Cd and Pb treatments, while they increased in roots under the treatments except for HCdLPb. In the leaves, the change patterns of the free proline in A7 and A64 were similar except for HCdHPb (Fig. 5). Compared with control, HCdHPb increased the free proline content in A7, while reduced the free proline content in A64. The higher free proline content was observed in the roots of A7 in LCdLPb as compared with control, while other treatments decreased the free proline contents in the roots of A7 (Fig. 5). The free proline contents in the roots of A64 presented a contrary trend. The soluble protein contents decreased in the leaves of both A7 and A64 under the combined stress of Cd and Pb when compared with the control (Fig. 5). On the contrary, they demonstrated higher contents in the roots.

As shown in Fig. 6, the activities of POD and CAT in leaves and roots of both A7 and A64 exhibited a significant decrease compared with control. POD and CAT activities

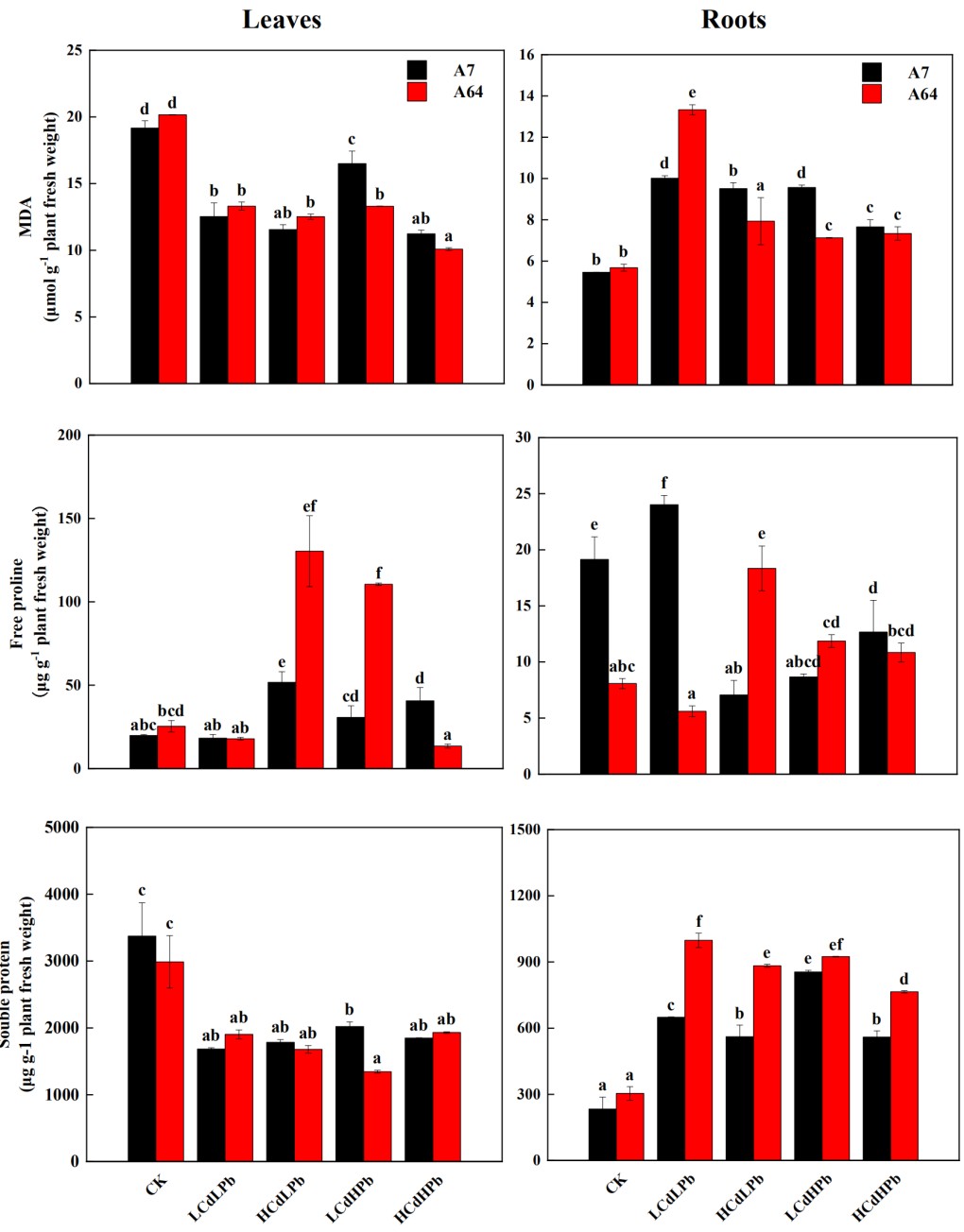

**Figure 5  MDA, free proline and soluble protein concentrations in leaves and roots of *S. matsudana* 'J172' (A7) and *S. matsudana* 'Yankang1' (A64) treated with different heavy metal concentrations.** CK, LCdLPb, HCdLPb, LCdHPb and HCdHPb present no added metals, $15 \mu M$ $CdCl_2$+ $250 \mu M$ $PbCl_2$, $30$ $\mu M$ $CdCl_2$+ $250 \mu M$ $PbCl_2$, $15 \mu M$ $CdCl_2$+ $500 \mu M$ $PbCl_2$ and $30 \mu M$ $CdCl_2$+ $500 \mu M$ $PbCl_2$. Each value represents the mean of the four plants ± SD. Different lowercase letters indicate significant differences among different treatments at the $P < 0.05$ level, as determined by LSD test.

had no significant differences between two willow clones. All the treatments significantly increased the activities of SOD in leaves and roots of both clones ($P < 0.05$). In addition,

the enzyme activities changed with the concentrations of combined stress of Cd and Pb, and the activities of POD and CAT in both leaves and roots were the lowest under HCdHPb treatment.

## Plant bioaccumulation under combined stress of Cd and Pb

Cd and Pb contents in leaf, stem, and root were presented in Table 1. HMs contents in the control treatment were lower than the limit of quantity (LOQ) ($LOQ_{Cd}$ <0.5 mg kg$^{-1}$, $LOQ_{Pb}$ <2.4 mg kg$^{-1}$). Pb mainly accumulated in the roots of A7 and A64, while the concentrations in leaves for Cd were higher on average than in other parts of the two willows. The augmentation of Pb in the solution markedly decreased the Cd accumulation in the leaves of two willow clones under the treatments of high Cd concentration, but significantly increased Cd concentrations in roots of A7. The increase of Cd concentration in the solution resulted in a significant increase of Pb contents in the shoots (leaves and stems) of A7 regardless of the Pb concentrations. The concentrations of Pb in stems of A64 were consistent with those of A7, but opposite in leaves.

BC$F_{Cd}$ was within the range of 0.16−0.25 and BC$F_{Pb}$ varied between 0.05−0.11 for these two willow clones (Table 2). Generally, BC$F_{Cd}$ values decreased with increasing Cd concentrations in the cultivated solution except for A7 exposed to high Pb levels. Similarly, BC$F_{Pb}$ decreased with the increasing Pb concentrations. T$F_{Cd}$ and T$F_{Pb}$ in two willow clones were lower than 1.0 under all treatments. It is worth noting that T$F_{Cd}$ was remarkably higher than T$F_{Pb}$. The TF'$_{Cd}$ exceeded 15 in all treatments for the two *S. matsudana* clones, and even arrived at 57 for A7 in HCdHPb. As for TF'$_{Pb}$, it is greater than 1.0 only in the HCdHPb for these two willow clones. The Ti of the two clones displayed a decrease trend with the increasing HMs concentrations. The Ti of both A7 and A64 was higher than 80%, even over 90% at the lowest Cd and Pb concentration. The Ti of these two clones did not show obvious difference.

## Correlation between antioxidant responses and metal tolerance capacity of willow clones under combined Cd and Pb stress

The relationships between the antioxidant responses and metal tolerance capacities of leaves and roots for A7 and A64 were studied by the Mantel test (Fig. 7). Ti had no significant correlation with antioxidant substances both in A7 and A64. The relationship between metal phytoextraction abilities also differed among willow clones. BC$F_{Cd}$ was significantly related to POD and CAT in leaves and roots of A7 ($P < 0.05$). However, BC$F_{Cd}$ had no significant relationship with POD and CAT in leaves and roots of A64, and it was related to SOD ($P < 0.05$). TF'$_{Cd}$ was significantly related to SOD in organs of A7 and A64 ($P < 0.05$), and the parameter was related with POD and CAT in organs of A64 ($P < 0.05$). Although T$F_{Pb}$ had no significant relationship with antioxidant substances in organs of A7, it had significant relationship with POD and CAT in organs of A64 ($P < 0.05$). Metal phytoextraction abilities of willow leaves and roots had different relationships with antioxidant responses under different combined stress of Cd and Pb. T$F_{Cd}$ and TF'$_{Pb}$ were significantly related with soluble protein and MDA in roots of A7 ($P < 0.05$), respectively, while they had no significant relationship with antioxidant substances in leaves of A7. For

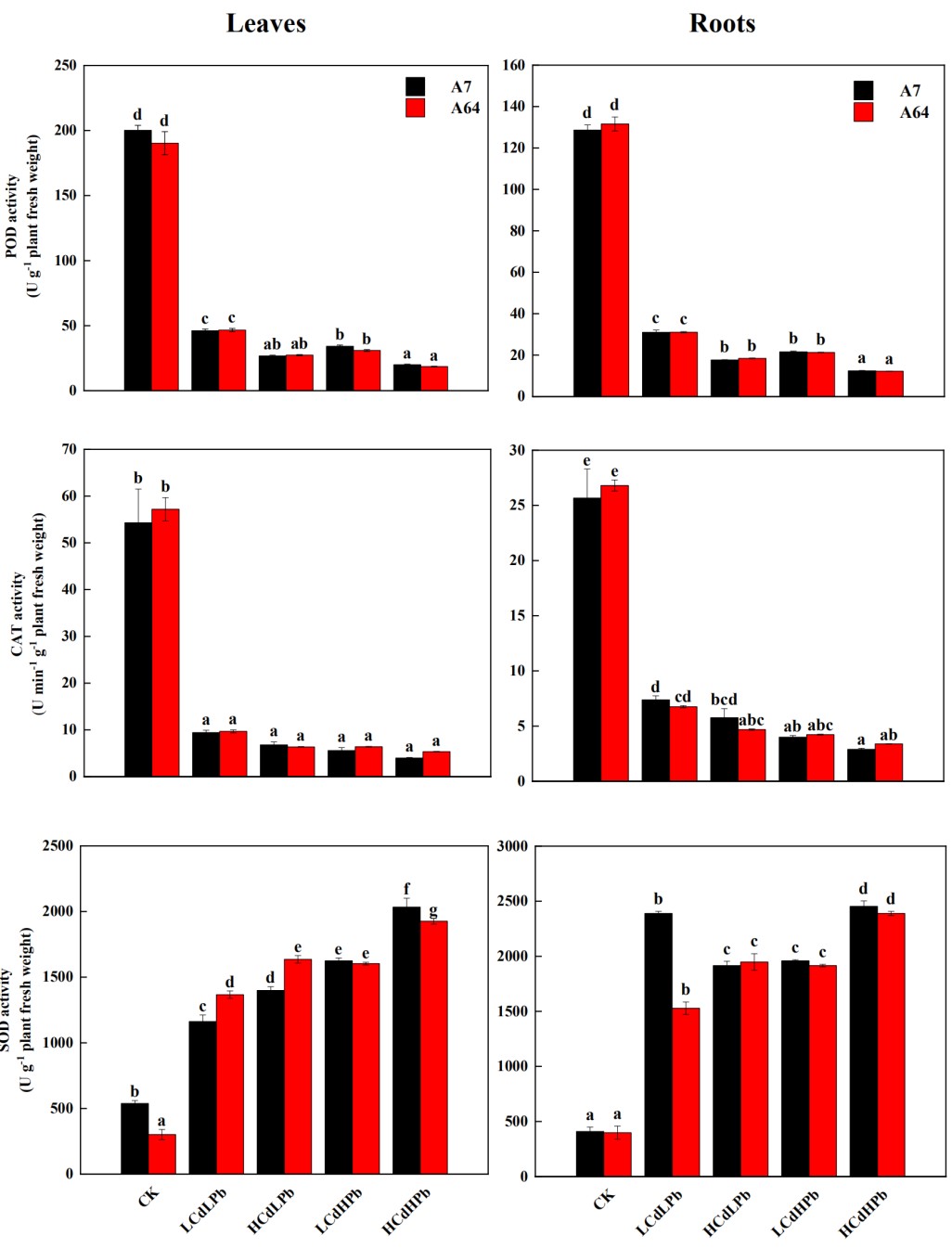

**Figure 6** Peroxidase (POD), catalase (CAT) and superoxide dismutase (SOD) activities in leaves and roots of *S. matsudana* 'J172' (A7) and *S. matsudana* 'Yankang1' (A64) treated with Cd and Pb. CK, LCdLPb, HCdLPb, LCdHPb and HCdHPb present no added metals, 15 $\mu$M CdCl$_2$+ 250 $\mu$M PbCl$_2$, 30 $\mu$M CdCl$_2$+ 250 $\mu$M PbCl$_2$, 15 $\mu$M CdCl$_2$+ 500 $\mu$M PbCl$_2$ and 30 $\mu$M CdCl$_2$+ 500 $\mu$M PbCl$_2$. Each value represents the mean of the four plants ± SD. Different lowercase letters indicate significant differences among different treatments at the $P < 0.05$ level, as determined by LSD test.

the A64, BC $F_{Pb}$ was related with soluble protein in leaves ($P < 0.05$), while the parameter had no significant relationship with antioxidant substances in roots. The responses of willow to Cd and Pb stresses were associated with both species and plant organs.

## DISCUSSION

### Growth and photosynthesis of two willow clones under combined Cd and Pb stress

Photosynthesis is an important index to study the responses of plants to environmental changes and the sensitivity to HMs, especially to high metal concentrations. The consistent decrease of *Gs* and *Tr* suggested that A7 and A64 closed the stomata under the treatments of higher Pb concentrations (LCdHPb, HCdHPb), which led to the decrease of transpiration rate and transpirational pull from the leaves. This result was similar to the Pb stress in *Silene viscidula* (*Wang et al., 2021*).

Notably, in this study, *Pn* of the two willow clones under all treatments was higher than that of the control, which was contrary to the previous results that HMs inhibited the *Pn* by affecting chloroplast morphology and the activity of photosynthesis-related enzymes (*Mishra et al., 2006*; *Noor et al., 2018*). $Cd^{2+}$ can reduce ribulose 1,5-bisphosphate carboxylase (RuBPC) activity by replacing $Mg^{2+}$ (*Zhu et al., 2021*), and $Pb^{2+}$ also inhibits enzyme synthesis of RuBPC (*Mishra et al., 2006*). We speculated that the competition between $Cd^{2+}$ and $Pb^{2+}$ avoided the substitution of cofactor-$Mg^{2+}$ and alleviated the damage to the activity of $CO_2$ fixation enzymes (Haider et al., 2021). *Qin & Song (2010)* found that the combination of $Pb^{2+}$, $Zn^{2+}$, $Cu^{2+}$ and $Cd^{2+}$ can promote the absorption of $Fe^{2+}$ and $Ca^{2+}$ and presented their high concentrations in roots. Thus, it may be the possible reason of the high photosynthesis that the co-occurrence of $Cd^{2+}$ and $Pb^{2+}$ might prevent them from competing with $Ca^{2+}$ and $K^+$ in ion channels of root and improve nutrients uptake from solution (*Amari, Ghnaya & Abdelly, 2017*). Previous studies have proved that combined stress of HMs can damage mitochondrial structure and disrupt plant respiration (*Li et al., 2021a*; *Li et al., 2021b*). *Lanier et al. (2019)* also found that the joint toxicity of $Cd^{2+}$ and $Pb^{2+}$ can lead to DNA breakage. These may explain that combined Cd and Pb treatments in the current study decreased the biomass of the two willow clones, although their photosynthetic rates were stimulated. The inhibition of combined $Cd^{2+}$ and $Pb^{2+}$ on plant growth may be related to other metabolic processes besides photosynthesis.

### Protection mechanisms of two willow clones against Cd and Pb

The production of ROS could cause imbalance of redox homeostasis, cellular damage and lipid peroxidation (*Arsenov et al., 2017*). MDA, an indicative product of peroxidation of membrane lipids, significantly decreased in the leaves of A7 and A64 under HMs treatments (Fig. 5), indicating the Cd-Pb stress had little damage to the membrane lipid of leaves. Similarly, *Redovnikovic et al. (2017)* found that MDA contents decreased in the leaves of *Populus nigra* 'Italica' after 4 months of exposure to the soil contaminated by Cd and Pb. Meanwhile, the *Pn* in the leaves of the two willow clones increased. It may be the possible defense mechanism that photosynthesis could alleviate oxidative stress induced by Cd and Pb in the leaves. MDA content increased in the roots of A7 and A64. The possible

**Table 1  Cd and Pb concentration in different plant parts of *S. matsudana* 'J172' (A7) and *S. matsudana* 'Yankang1' (A64) treated with different Cd and Pb concentrations.** CK, LCdLPb, HCdLPb, LCdHPb and HCdHPb present no added metals, 15 $\mu$M CdCl$_2$ + 250 $\mu$M PbCl$_2$, 30 $\mu$M CdCl$_2$ + 250 $\mu$M PbCl$_2$, 15 $\mu$M CdCl$_2$ + 500 $\mu$M PbCl$_2$ and 30 $\mu$M CdCl$_2$ + 500 $\mu$M PbCl$_2$. Each value represents the mean of the four plants SD. Different letters within the same species indicate significant differences among four treatments. Asterisks (*) denote a significant difference between the *S. matsudana* 'J172' and *S. matsudana* 'Yankang1' in same heavy metal treatments at the *P* < 0.05 level, as determined by LSD test.

| Willow specie | Treatment | Cd contents ($\mu$g g$^{-1}$ dry weight) | | | Pb contents ($\mu$g g$^{-1}$ dry weight) | | |
|---|---|---|---|---|---|---|---|
| | | Leaf | Stem | Root | Leaf | Stem | Root |
| *S. matsudana* 'J172' (A7) | LCdLPb | 120.76 ± 19.84a* | 52.86 ± 7.22a | 128.61 ± 1.93a | 224.28 ± 46.36a* | 182.27 ± 1.96a* | 22596.87 ± 837.01a* |
| | HCdLPb | 357.89 ± 3808c* | 91.38 ± 10.65b | 145.18 ± 1.48b* | 343.69 ± 40.08a | 364.69 ± 9.97b | 18966.97 ± 3030.19a |
| | LCdHPb | 129.08 ± 28.81a* | 44.38 ± 0.20a | 132.24 ± 3.76a | 265.40 ± 26.59a* | 225.51 ± 36.37a* | 30100.96 ± 1310.64b |
| | HCdHPb | 220.34 ± 13.91b | 96.91 ± 11.46b | 157.26 ± 0.94c* | 514.19 ± 66.03b* | 644.34 ± 20.32c* | 31878.12 ± 1463.12b* |
| *S. matsudana* 'Yankang1' (A64) | LCdLPb | 225.04 ± 8.37b* | 47.66 ± 5.56a | 132.40 ± 4.04a | 524.13 ± 30.70c* | 43.53 ± 2.48a* | 27611.26 ± 947.20b* |
| | HCdLPb | 233.12 ± 25.46b* | 108.35 ± 4.53c | 176.09 ± 3.50b* | 328.40 ± 45.50ab | 355.41 ± 20.13b | 19216.61 ± 1676.94a |
| | LCdHPb | 236.49 ± 31.95b* | 55.68 ± 3.85a | 126.31 ± 2.21a | 421.68 ± 67.77bc* | 663.75 ± 16.01c* | 27674.92 ± 276.04b |
| | HCdHPb | 150.96 ± 13.26a | 82.93 ± 3.83b | 167.71 ± 3.79b* | 236.90 ± 8.01a* | 932.24 ± 15.50d* | 36978.99 ± 1508.99c* |

**Table 2 Ti, BCF, TF and TF' of *S. matsudana* 'J172' (A7) and *S. matsudana* 'Yankang1' (A64) treated with different combined Cd and Pb concentrations.** CK, LCdLPb, HCdLPb, LCdHPb and HCdHPb present no added metals, 15 μM CdCl2 + 250 μM PbCl2, 30 μM CdCl2 + 250 μM PbCl2, 15 μM CdCl2 + 500 μM PbCl2 and 30 μM CdCl2 + 500 μM PbCl2. Each value represents the mean of the four plants ± SD. Different letters within the same species indicate significant differences among four treatments. Asterisks (*) denote a significant difference between *S. matsudana* 'J172' and *S. matsudana* 'Yankang1' in same heavy metal treatments at the $P < 0.05$ level, as determined by LSD test.

| Genotypes | Treatments | Ti (%) | BCF Cd | BCF Pb | TF Cd | TF Pb | TF' Cd | TF' Pb |
|---|---|---|---|---|---|---|---|---|
| *S. matsudana* 'J172' (A7) | LCdLPb | 96.88 ± 2.53a | 0.223 ± 0.019c | 0.066 ± 0.004b* | 0.439 ± 0.053a | 0.008 ± 0.000a* | 28.679 ± 1.643b* | 0.549 ± 0.072b* |
| | HCdLPb | 84.82 ± 4.92a | 0.178 ± 0.015a* | 0.092 ± 0.009c* | 0.713 ± 0.076b | 0.020 ± 0.002b | 27.792 ± 3.040b | 0.780 ± 0.107b |
| | LCdHPb | 88.61 ± 2.28a | 0.178 ± 0.002a* | 0.050 ± 0.006a* | 0.363 ± 0.007a* | 0.008 ± 0.001a* | 17.421 ± 2.010a | 0.353 ± 0.037a |
| | HCdHPb | 89.33 ± 8.90a | 0.184 ± 0.005b | 0.058 ± 0.002ab* | 0.655 ± 0.061b* | 0.020 ± 0.000b* | 57.263 ± 4.957c* | 1.759 ± 0.076c* |
| *S. matsudana* 'Yankang1'(A64) | LCdLPb | 91.11 ± 4.77a | 0.230 ± 0.008b | 0.105 ± 0.003b* | 0.434 ± 0.032a* | 0.003 ± 0.000a* | 15.301 ± 0.898a* | 0.090 ± 0.002a* |
| | HCdLPb | 87.60 ± 4.91a | 0.220 ± 0.001b* | 0.111 ± 0.007b* | 0.661 ± 0.021b | 0.018 ± 0.001b | 21.645 ± 0.814ab | 0.608 ± 0.048b |
| | LCdHPb | 82.04 ± 1.85a | 0.245 ± 0.007b* | 0.081 ± 0.004a* | 0.543 ± 0.032ab* | 0.023 ± 0.000c* | 18.567 ± 1.641a | 0.800 ± 0.061b |
| | HCdHPb | 86.84 ± 3.48a | 0.161 ± 0.001a | 0.096 ± 0.003ab* | 0.511 ± 0.029a* | 0.024 ± 0.001c* | 26.552 ± 2.478b* | 1.284 ± 0.153c* |

**Notes.**

Ti, Tolerance index; BCF, Bioconcentration factor; TF, Translocation factors for concentration; TF', Translocation factors for accumulation.

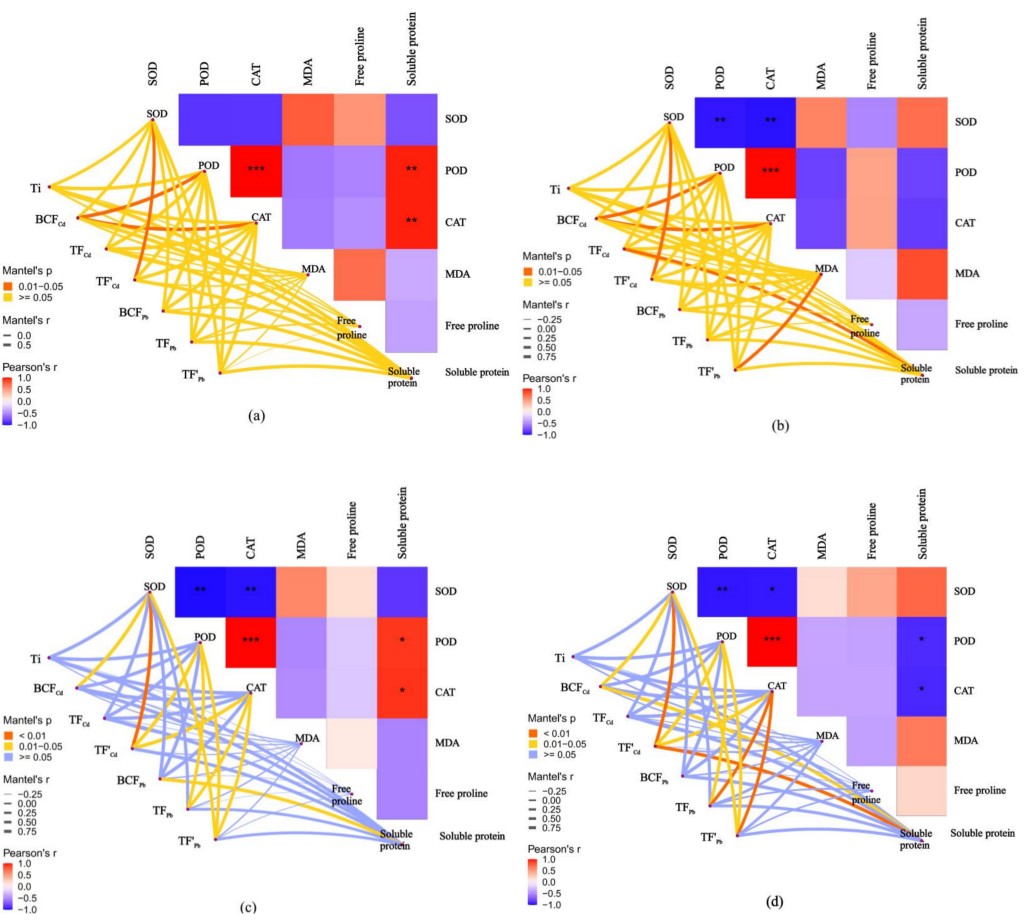

**Figure 7** Correlation analysis illustrating the relationship between antioxidant system in different parts and Ti, BCF, TF and TF' of two clonesin treatments with combined Cd and Pb. (A) A7 leaves; (B) A7 roots; (C) A64 leaves; (D) A64 roots. A color gradient denotes the Pearson's correlation coefficients, and red indicates positive correlations, blue indicates negative correlation. * $P < 0.05$; ** $P < 0.01$; *** $P < 0.001$. The edge color denotes the statistical significance.

reason is that they could not carry out photosynthesis due to the absence of photosynthetic chloroplasts, which caused lipid peroxidation immediately as indicated by an increase in the MDA content (*Lei, Korpelainen & Li, 2007*).

Free proline is the critical indicator of plant tolerance capacity to stress, such as HMs, drought, and salinity (*Szabados & Savoure, 2010*). In this study, the free proline increased in the leaves of two willow clones especially A64 under HCdLPb and LCdHPb (Fig. 5). These results showed that A64 has a higher tolerance than A7 when the concentration of one heavy metal is higher in the combined treatments. *Arsenov et al. (2017)* also found the content of free proline in willows grown in Cd-contaminated soil varied with tree species. In addition, higher free proline was observed in the leaves which may be due to its protective role in plant metabolism.

The presence of Cd and Pb reduced soluble protein contents in the leaves of A7 and A64, which illustrated HMs weakened leaf metabolic activity and osmotic pressure

regulation (*Xie & Wang, 2021*). The elevated soluble protein content in the roots indicated their tolerance capacity to HMs. The possible reason was that soluble proteins in the aboveground parts are transported to the underground parts to protect the roots which were directly exposed to HMs. Soluble protein is assumed to be one of the mechanisms of HMs tolerance and is related with plant species (*Zheng, Fei & Huang, 2009*). The soluble protein content in roots of A64 was significantly higher than that of A7, which is similar with the study of *Yan (2017)* that Pb had different effects on soluble protein contents in leaves of three shrubs. The enhanced soluble protein contents related to the enhanced tolerance to HMs (*Zheng, Fei & Huang, 2009*), and thus we further demonstrated that the clone A64 has a higher tolerance than A7.

The previous study found that plants have complex ROS scavenging mechanisms to decrease cellular oxidative damage and increase resistance to HMs (*Rastgoo & Alemzadeh, 2011*). In the present study, the activities of POD and CAT significantly decreased under all treatments (Fig. 6), which could lead to the decline in the tolerance under the severe HMs stress (*Li et al., 2012*). The formation of protein complex with metals could change the structure or integrity of proteins, which may be the reason that the enzyme activities decreased (*Hou et al., 2007*). However, SOD activities were significantly higher than those in the control (Fig. 6), indicating that higher SOD activity could alleviate the oxidative damage of plant organs when exposed to HMs. These enzymes were located at different cellular sites, which had different resistances to heavy metals (*Hou et al., 2007*).

## Potential Cd and Pb accumulation capacity of two willow clones

Our study found that Pb accumulated mainly in the roots of both willow clones (Table 1). *Vandecasteele et al. (2005)* also demonstrated that *Salix fragilis* L. 'Belgisch Rood' and *S. viminalis* L. 'Aage' are root accumulator for Pb. The possible reason was that the Pb absorbed from the solution was deposited on the root tip surface first and the existence of the Kessler band hindered its migration (*Peng, Wang & Wu, 1989*). Cd in A64 was transported faster than Pb and accumulated in leaves through the symplast pathway (*Guo et al., 2021*; *Peng, Wang & Wu, 1989*). However, Cd accumulated in the roots of A64 in the treatment of highest concentrations of Cd and Pb. *Nishizono, Ichikawa & Suzuki (1987)* also found 70%–90% of Cd was gathered on the cell wall of the root tip of the hyperaccumulator plant *Athyrium Yokoscense*. Plants can prevent potentially toxinc elements, such as Cd, from being translocated aboveground by exclusion mechanism (*Zuzolo et al., 2022*). Cd deposition on the cell wall prevented $Cd^{2+}$ from entering the cell and transporting to the shoots. In addition, Cd in A7 was transported into the leaves from the roots in HCdHPb. The distribution of Cd also reveals that two willow clones may have different transport mechanisms for the same metal. Plants are able to extract heavy metal elements and respond to the excess of toxic chemicals, so it is important to screen the plant species for enhancing the phytoremediation performance (*Zuzolo et al., 2022*). The further mechanism is in need to explore by the molecular study.

HMs accumulation potential of the plant is the critical parameter for the plant screening in the phytoremediation of the HMs contaminated sites (*Lin et al., 2020*). The BCF value is a measure of plants' ability to remove HMs from the environment and the BCF >1

illustrated that the plant has a high accumulation capacity (*Huang et al., 2019*). The BCFs of the two willow clones were lower than 1.0 regardless of the concentrations of Cd and Pb, indicating they are not hyperaccumulators. However, *Salix psammophila* has a high accumulation capacity of Cd, which might be caused by the different metabolic responses of willow species (*Li et al., 2021a*; *Li et al., 2021b*).

Low TF values under all the treatments indicated that Cd and Pb translocation from roots to shoots are weak for both willow clones. The values of $TF_{Cd}$ and $TF'_{Cd}$ implied that *S. matsudana* displayed relatively higher phytoremediation efficiency in heavy Cd-contaminated waters compared with that in low level Cd-contaminated waters. Compared with $TF_{Pb}$ of A7 in HCdHPb, A64 showed a better ability to extract the Pb from root to the aboveground tissue than A7. Thus, A64 demonstrated the high capacity to extract Pb from the high combined concentrations of Pb and Cd. The BCF, TF and TF' of Cd and Pb for the two new *S. matsudana* clones confirmed that the phytoextraction of Cd or Pb will be affected by another heavy metal. The tolerance to metal is a key prerequisite for metal accumulation and phytoremediation (*Ali, Khan & Sajad, 2013*).

According to the threshold proposed by *Lux et al. (2004)*, these two willow clones in this study can be defined as high tolerance (Ti >60%). Root system was also an important mechanism for *Triarrhena sacchariflora* to grow in waters polluted by Cd and Pb (*Xin, Zhang & Tian, 2018*). Plants can accumulate metals in their tissues by triggering the mechanism of detoxification. Thus, the combined exposure of HMs can be complex with proteins in the roots to form stable chelates and remain in the root after HMs were absorbed by roots. It is one of the important mechanisms to reduce the toxicity of HMs to plants.

## CONCLUSIONS

The antioxidant responses of *S. matsudana* 'J172' and *S. matsudana* 'Yankang1' to combined stress of Cd and Pb were different between aboveground and underground tissues. Photosynthesis played important roles in the resistance of Cd and Pb toxicity, and thus decreased the contents of MDA in leaves. Soluble protein accumulated in the roots of willow to alleviate the damage caused by Cd and Pb, and it was higher in the roots of A64 than that of A7. A64 has a higher tolerance than A7. The removal of ROS by SOD is the important mechanism in these two willow clones under combined Cd and Pb stress. A7 and A64 are good candidates for phytoremediation of Cd and Pb in the combined contamination environment. The tolerance and detoxification mechanism of willows is very complex, thus further studies are needed to investigate for the application of *Salix spp.* clones in HMs contaminated environment. To conclude, the two *Salix spp.* clones are good candidates in the phytoremediation of environments polluted by various HMs.

## ACKNOWLEDGEMENTS

We are grateful to the Shandong Academy of Forestry for providing the two new clones, Salix matsudana 'J172' (A7) and Salix matsudana 'Yankang1' (A64) for this study.

### Funding

This work was supported by the National Natural Science Foundation of China (grant numbers 31870606; 41877424; 32071559); the Research Leader Studio Project (grant number 2021GXRC094); the Key R & D project of Shandong Province (2021LZGC005-02), and the Fundamental Research Funds for the Central Universities, CHD (300102351505). The funders had no role in study design, data collection and analysis, decision to publish, or preparation of the manuscript.

### Grant Disclosures

The following grant information was disclosed by the authors:
National Natural Science Foundation of China: 31870606, 41877424, 32071559.
Research Leader Studio Project: 2021GXRC094.
Key R & D project of Shandong Province: 2021LZGC005-02.
Fundamental Research Funds for the Central Universities, CHD: 300102351505.

### Competing Interests

The authors declare there are no competing interests.

### Author Contributions

- Chuanfeng Zhang conceived and designed the experiments, performed the experiments, analyzed the data, prepared figures and/or tables, authored or reviewed drafts of the article, and approved the final draft.
- Baoshan Yang conceived and designed the experiments, performed the experiments, authored or reviewed drafts of the article, and approved the final draft.
- Hui Wang conceived and designed the experiments, analyzed the data, prepared figures and/or tables, authored or reviewed drafts of the article, and approved the final draft.
- Xiaohan Xu conceived and designed the experiments, performed the experiments, analyzed the data, prepared figures and/or tables, and approved the final draft.
- Jiaxing Shi conceived and designed the experiments, prepared figures and/or tables, and approved the final draft.
- Guanghua Qin performed the experiments, prepared figures and/or tables, and approved the final draft.

### Data Availability

The raw measurements are available in the Supplemental Files.

### Supplemental Information

Supplemental information for this article can be found online at http://dx.doi.org/10.7717/peerj.14521#supplemental-information.

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
