# Peer review of "Metal tolerance capacity and antioxidant responses of new Salix spp. clones in a combined Cd-Pb polluted system"

_PeerJ, doi:10.7717/peerj.14521_

## Round 0.1 · original submission · Major Revisions

Please respond to all the comments. Additionally, 1) you are requested to include pictures of Salix spp. clones representing the phenotypic response to control and HMs application, 2) Please give the reason for application of HMs at 3 weeks old plants and for a period of 35 days, 3) Please give more reasoning for the selection of Salix spp. clones for the current study and if possible, make a comparison with already known clones

·

Basic reporting

The manuscript entitled ‘Metal tolerance capacity and antioxidant responses of new Salix spp. clones in a combined Cd-Pb polluted system’ has investigated the willow spp tolerance capacity under heavy metals. Authors have taken two heavy metals Cd and Pb into account. Pb is highly toxic and its need of time to find new clones for its remediation. The topic is consistant with the journal aims and scopes. In general, the underlying data offers the possibility for the use of Salix spp for phytoremediation of Cd and Pb in the contaminated soil. I have some comments that the authors could consider.

Experimental design

well clear

Validity of the findings

no comments

Additional comments

1. How authors choose the concentration of Cd and Pb, Is there any specific reason to choose these concentration
2. Did authors have any idea about biomass parameters of Salix spp alone in Cd or Pb. Comparison between combined heavy metals and alone would be helpful to better understand.
3. Authors are unable to find correlation between concentration of HMs and free proline content as in fig 4

·

Basic reporting

Abstract part:
Adequate in length; good summary of the study done
line 24: abbreviations in abstract have to be not used
line 27: sentence not clear; should be reformulated
lines 27 & 28: contents - concentration, i suppose?

Introduction:
Clear and fully informative
line 78: some words necessary to explain the choice of the experimental design: why with a hydroponic system? advantages & drawbacks

Experimental design

Mat & Meth part:
Many details missing!
About the experimental design: you have to justify the Cd and Pb concentrations used ; did you measure periodically the HM concentrations on the hydroponic bath?
line 91: pure water without oxygen added?
line 95: criteria which defined uniformity should be defined
line 121: material (and conditions) used to pulverize plant samples have to be specified
line 143: is rather “statistical”

Validity of the findings

Results part:
no remarks on results; briefly and correctly described; no original data really!
line 159: abbreviation used should be redefined!

Discussion part:
clearly written & supported by relevant references

Conclusions part:
good summary of the main results but with no perspectives?

Fig. & Table
For all captions: codes used to define modalities tested (on X-axis) should be indicated
Fig. 2: caption not clear; should be developed to better understand the figure
Fig. 3: figures too small; difficult to read!
Fig. 5: same remark
Fig. 6: caption has to be completed to explain the colour code
Table 1: codes used to define modalities (col. 2: "treatments") tested should be indicated in the caption

Additional comments

In my opinion, the article is moderately interesting because it has no innovative character.

Reviewer 3 ·

Basic reporting

Understandable text, as well as experimental intent

Experimental design

No comment

Validity of the findings

The results elevate the two salix clones as potential tools for phytoextraction in hydroponics.
It lays the foundation for further future studies for the full understanding of the mechanisms activated in phytoextraction.

Annotated reviews are not available for download in order to protect the identity of reviewers who chose to remain anonymous.

---

## Round 0.2 · Major Revisions

Keeping in view the comments of reviewers, we are returning the manuscript for you to incorporate the following suggestions from the Section Editor:

> Conclusions are not supported by the data.

> Specifically "The photosynthesis and free proline play important roles in the resistance of Cd and Pb toxicity in A7 and A64" (conclusion) and "The concentrations of free proline and soluble protein were obviously higher in A64 than A7, and the two substances played important roles in the tolerance of A64 to Cd and Pb"(abstract).

> While photosynthesis, free proline, and soluble protein change in response to treatment, this manuscript does not test whether or not these changes are important for resistance in these lines.:

> +++ "The tolerance index was more than for both clones under all treatments, indicating they have high tolerances for combined stress of Cd and Pb". Without comparison to poor performing clones it is hard to put this in context. Maybe the treatments used here just were not very severe and most clones would show similar tolerance

> +++ Also needs careful editing for grammatical errors. Just a few examples: line 26 " demonstrating photosynthesis may be one of the metabolic processes to resist Cd-Pb stress" should be " demonstrating that photosynthesis may be one of the metabolic processes used to resist Cd-Pb stress"

> +++ line 124 "To compare the restraint of combined Cd and Pb stress to the biomass of new breeding willow" I am not sure what is meant here but maybe "To compare the resistance of new breeding willow clones to the combined stress of Cd and Pb..."

> +++ Figure 7 legend "treated" --> "treatment" ; "bule" --> "blue". etc.

> +++ These are just a few examples, the whole manuscript needs to be carefully checked.


·

Basic reporting

Authors have incorporated all the suggestions from m side.
I think this article is publishable in it present form.

Experimental design

no comments

Validity of the findings

no comment

Additional comments

no comment

---

## Round 0.3 · accepted · Accept

In my opinion, the authors have significantly improved the manuscript in light of reviewer comments. It may be processed to the next step.